# Out-of-Season Influenza during a COVID-19 Void in the State of Rio de Janeiro, Brazil: Temperature Matters

**DOI:** 10.3390/vaccines10050821

**Published:** 2022-05-23

**Authors:** Rohini Nott, Trevon L. Fuller, Patrícia Brasil, Karin Nielsen-Saines

**Affiliations:** 1Department of Pediatrics, David Geffen School of Medicine at University of California Los Angeles, Los Angeles, CA 90095, USA; knielsen@mednet.ucla.edu; 2Institute of the Environment & Sustainability, University of California Los Angeles, Los Angeles, CA 90095, USA; fullertl@g.ucla.edu; 3Laboratorio de Doenças Febris Agudas, Instituto Nacional de Infectologia, Fundação Oswaldo Cruz, Rio de Janeiro 21040-360, RJ, Brazil; patricia.brasil@ini.fiocruz.br

**Keywords:** COVID-19 pandemic, Rio de Janeiro influenza epidemic, climate change, replicative advantage, out-of-season influenza, influenza epidemic

## Abstract

An out-of-season H3N2 type A influenza epidemic occurred in the State of Rio de Janeiro, Brazil during October–November 2021, in between the Delta and Omicron SARS-CoV-2 surges, which occurred in July–October 2021 and January–April 2022, respectively. We assessed the contribution of climate change and influenza immunization coverage in this unique, little publicized phenomenon. State weather patterns during the influenza epidemic were significantly different from the five preceding years, matching typical winter temperatures, associated with the out-of-season influenza. We also found a mismatch between influenza vaccine strains used in the winter of 2021 (trivalent vaccine with two type A strains (Victoria/2570/2019 H1N1, Hong Kong/2671/2019 H3N2) and one type B strain (Washington/02/2019, wild type) and the circulating influenza strain responsible for the epidemic (H3N2 Darwin type A influenza strain). In addition, in 2021, there was poor influenza vaccine coverage with only 56% of the population over 6 months old immunized. Amid the COVID-19 pandemic, we should be prepared for out-of-season outbreaks of other respiratory viruses in periods of COVID-19 remission, which underscore novel disease dynamics in the pandemic era. The availability of year-round influenza vaccines could help avoid unnecessary morbidity and mortality given that antibodies rapidly wane. Moreover, this would enable unimmunized individuals to have additional opportunities to vaccinate during out-of-season outbreaks.

## 1. Introduction

A horrific case load of COVID-19 cases swept across the world and across Brazil in the South American winter of 2021 (June–September), leading to a pandemic surge among susceptible, un-boosted elderly individuals previously immunized with the inactivated virus vaccine, Coronovac (Sinovac Biotech/Butantan Institute, São Paulo, Brazil) [1]. This phenomenon was the subject of a number of modeling studies attempting to explain pandemic dynamics [2,3]. By spring 2021 (October–December), the outlook had improved considerably, with COVID-19 cases plummeting dramatically by mid-October [4]. By then, a considerable proportion of the population had been boosted by recent natural infection with the Delta strain of SARS-CoV-2 and/or received a booster dose with the Pfizer-BioNTech BNT162b2 (Pfizer-BioNTech, Kalamazoo, MI, USA) mRNA vaccine. By late spring (October–November 2021), the State of Rio de Janeiro (RJ), a region severely affected by the COVID-19 pandemic, was reporting zero hospital admissions due to COVID-19 [4]. This brief reprieve in respiratory virus infections was followed by an impactful and completely unexpected influenza H3N2 epidemic which peaked by late November 2021 [5] and declined throughout December 2021, disappearing by early January 2022 at which time the SARS-CoV-2 Omicron strain surged in RJ and throughout Brazil [6].

This little publicized out-of-season influenza epidemic occurred during a temporary COVID-19 void in between massive Delta and Omicron SARS-CoV-2 surges, which occurred in July–October 2021 and January–April 2022, respectively [7]. We evaluated whether climate change and poor influenza immunization coverage contributed to the genesis of this perfect storm and share findings based on a descriptive analysis of the data considered.

## 2. Materials and Methods

### 2.1. Calculation of Weekly SARS-CoV-2 Case Numbers

Data on the number of weekly confirmed state SARS-CoV-2 cases from the epidemiologic weeks of 4 July 2021 to 26 December 2021 were obtained from the RJ State Health Department [4]. Confirmed cases were reported to the Health Department based on positive molecular and/or antigen tests identified through the public network of the Brazilian Single Unified Health System (SUS) hospitals, clinics and ambulatory settings, public and private testing sites, private clinics, hospitals and pharmacies distributed throughout the state [4,6].

### 2.2. Calculation of Weekly Influenza Case Numbers

We calculated weekly state influenza case numbers based on weekly influenza hospitalizations reported to the RJ State Health Department from 4 July 2021 to 26 December 2021 [6] using CDC methodology [8]. Per the CDC method of estimating the number of influenza cases annually, flu hospitalizations comprise 1.3% of total cases and deaths 0.1% of influenza cases [8]. The influenza burden was estimated from CDC data from 2018–2019, in which for 29 million symptomatic cases, there were 380,000 hospitalizations and 28,000 deaths due to influenza.

### 2.3. SARS-CoV-2 Immunization Coverage

We assessed SARS-CoV-2 vaccination coverage in RJ, which represented the proportion of the state population as of 12 years of age who received either two doses of Sinovac-CoronaVac (Sinovac Biotech/Butantan Institute, São Paulo, Brazil), Oxford–AstraZeneca (AstraZeneca, Oxford, UK/Fiocruz, RJ, Brazil) or Pfizer-BioNTech BNT162b2 (Pfizer-BioNTech, Kalamazoo, MI, USA) mRNA vaccines or one dose of the Janssen vaccine (Janssen Biotech, Inc., Horsham, PA, USA), all provided by SUS, which is the only health network through which individuals in Brazil received COVID-19 immunizations. Data on SARS-CoV-2 immunization coverage was obtained from the RJ State Health Department [4]. Though the vaccine was only licensed for individuals over 12 years old, weekly SARS-CoV-2 case numbers represent those affected with COVID-19 among the entire state population.

### 2.4. Influenza Immunization Coverage

Information about influenza immunization coverage was obtained from the RJ State Health Department [6]. Flu vaccination in Brazil is recommended for all individuals as of 6 months of age [9]. Immunization coverage represented the proportion of individuals in RJ state who received one dose of influenza vaccine between May 2021 to August 2021, the annual period in which flu vaccines are provided for the state [9]. The vaccine administered in 2021 was a trivalent vaccine consisting of two type A influenza strains (Victoria/2570/2019 H1N1, Hong Kong/2671/2019 H3N2) and one type B strain (Washington/02/2019, wild type) [10].

### 2.5. Association between Out-of-Season Influenza Outbreak and Climatic Variations

To assess the possible association between the out-of-season influenza outbreak and climatic variations, we evaluated temperature measurements from ten representative weather stations distributed across the RJ state. These weather stations were: Arraial do Cabo, Campos dos Goytacazes, Cambuci, Duque de Caxias, Macaé, Pico do Couto, Resende, Rio de Janeiro, Seropédica and Valença. These data were available through the Brazilian National Institute of Meteorology [11] (Figure 1). Maximum and minimum temperatures in Celsius units recorded across RJ state for October and November from 2016 to 2021 were abstracted. For comparison of weather patterns over six years, a Wilcoxon signed rank test was performed comparing values for maximum and minimum temperatures for October and November in 2021 with the mean values for October and November for 2016 to 2020 for RJ state. A *p*-value less than 0.05 was considered statistically significant.

## 3. Results

Between 4 July 2021 to 26 December 2021, 234,621 cases of confirmed COVID-19 were reported to the RJ State Health Department, with the Delta variant being responsible for the majority of cases [4]. By 7 November 2021, COVID-19 cases had declined precipitously, coinciding with a rise in influenza cases (Figure 2). Cases of COVID-19 during this period accounted for 40,712 hospitalizations and 13,377 deaths, with a case fatality rate of 5.7 percent in the state during this time frame.

An estimated 68,077 cases of influenza due to the H3N2 Darwin type A influenza strain [6] occurred between November to December 2021 [9], with a case fatality rate of 0.1 percent. Unlike the influenza epidemics of previous years which occurred in May and June, the influenza epidemic in 2021 occurred in November. Furthermore, the circulating strain of H3N2 influenza virus was different from the H3N2 strain provided in the trivalent vaccine during the fall/winter period of 2021. Influenza vaccine coverage for the RJ state population during the spring/winter of 2021 was less than 60% [5]. This encompasses anyone as of 6 months of age. Influenza hospitalizations, cases, circulating strains and vaccine coverage for the State of Rio de Janeiro from 2016 to 2021 are shown in Table 1.

Interestingly, state weather patterns during October and November of 2021 (spring in the Southern Hemisphere) when the influenza virus epidemic occurred were distinctly different from the five preceding years. In October 2021, maximum temperatures were nearly 2 degrees Celsius less than the average of the maximum temperatures in the five preceding years, while minimum temperatures were nearly two degrees Celsius lower than the mean value for the five preceding years. The same phenomenon was observed for the month of November, although the difference in temperature magnitude was approximately 1 degree Celsius for maximum and minimum temperatures compared to the five preceding years. For both months, the change in temperature noted in 2021 was statistically significant (Figure 3). Indeed, the mean temperature during the influenza epidemic during October and November 2021 (20.4 °C and 21 °C, respectively) was similar to the mean temperatures reported during the winter months of July and August (20.1 and 21 °C, respectively in 2021) in a typical year [11,22]. The median and interquartile ranges of monthly minimum, mean and maximum temperatures of 17 weather stations in the State of Rio de Janeiro are reported from 2016 to 2021 in Table 2. In the preceding years, in the State of Rio de Janeiro, there were no periods in which a decrease in temperature outside of the winter season produced an increase in the cases of influenza. As shown in Table 2, however, in the five preceding years, temperatures were warmer than in 2021.

The general population of the State of RJ had poor vaccine coverage against influenza in the winter of 2021 (55.7% of the population over 6 months old was vaccinated) [5]. In addition, there was a mismatch between influenza virus vaccine strains in 2021 and the influenza virus which circulated in October/November 2021. Furthermore, a decline in COVID-19 cases following an epidemic Delta variant surge and higher COVID-19 immunization rates [4] led to a SARS-CoV-2 void in October/December 2021. During this void, an influenza epidemic occurred out-of-season in the State of RJ with an estimated 68,077 cases. A decline in mitigation measures might have contributed to the exacerbation of influenza cases in the spring/summer of 2021. Although there were appropriate conditions for the influenza virus to circulate in the winter of 2021, minimal to no influenza virus circulated simultaneously to the Delta SARS-CoV-2 epidemic surge during which 234,621 SARS-CoV-2 cases were reported. Similarly, when Omicron cases appeared in January 2022 leading to a COVID-19 pandemic surge, cases of the influenza virus plummeted across the state [4]. Nevertheless, the influenza epidemic migrated south and continued in the south and in other southeast areas of Brazil during January 2022, until the Omicron variant surged nationally. At that point the number of influenza cases declined precipitously across the country.

## 4. Discussion

Early in the COVID-19 pandemic, reports described dramatic reductions in circulating influenza and RSV strains globally [23,24,25], with one report even considering the extinction of specific influenza viral lineages [26]. As the pandemic progressed, other respiratory viruses were shown to circulate off-season, frequently during a period of SARS-CoV-2 decline, a phenomenon likely driven by changing weather patterns, other prevailing competing viruses in the ecosystem and possibly limited use of mitigating measures such as masks and social distancing [27]. One such example is respiratory syncytial virus, which occurred off-season in the summer of 2021 in the United States, Japan and Australia, also coinciding with a low COVID-19 case load [28,29,30]. In South Africa, a similar phenomenon was observed where influenza circulated off-season [31].

Typically, the seasonality of influenza in Brazil is characterized by a biannual pattern, with a larger peak in the winter month of June and a smaller peak during the summer month of January [32]. Our findings showed a seasonally unexpected peak in influenza cases in November 2021, a period in which there was a low COVID-19 case load. Compared to the influenza epidemics in previous years that occurred in May and June, the 2021 influenza epidemic in November was unanticipated. We found that there was a distinctly lower mean temperature than expected for the month of October and November based on data from 2016 to 2020, which may have contributed to the off-season influenza epidemic.

In our analysis, a void in the circulating prevailing pandemic, a mismatch between the circulating influenza strain and influenza vaccination and climate changes fostering improved weather conditions for flu viral spread may be predictive of off-season influenza virus circulation. Initially in the COVID-19 pandemic, interventions mitigating viral spread such as masks and social distancing likely prevented the circulation of other viruses. However, a combination of reduced mitigating measures during the brief Rio COVID-19 void period, coupled with colder weather, favoring population clustering indoors may have played an important role in the out-of-season Rio flu outbreak. Studies have suggested that most person-to-person transmission events occur indoors, perhaps driven by the indoor climate, proximity and limited amount of breathing air [33,34,35]. During cold weather, people are more likely to be confined to indoor spaces, which can facilitate virus transmission [36]. Furthermore, during changing weather conditions, people may have decreased host immune defense mechanisms, which make them susceptible to illness [36]. In contrast, by mid-November, the COVID-19 vaccination rate in RJ exceeded 70 percent (as shown in Figure 2), which may have played a role in protecting the State of RJ population from the Delta strain. In addition, a considerable proportion of the population acquired COVID-19 during the Delta surge before November 2021. Thus, both vaccination and recent natural immunity likely contributed to a lack of resurgence of COVID-19 cases.

In the specific case of influenza, when there are available vaccines for prevention and oral antiviral treatment to abort disease progression, year-round influenza vaccines and antiviral agents would help avoid unnecessary morbidity and mortality, given that antibodies rapidly wane and are not protective for an entire year. Furthermore, while we live through cycles of COVID-19 surges during the pandemic era, we should be prepared for off-season outbreaks of other respiratory viruses which can potentially prevail unexpectedly during periods of COVID-19 remission, as underscored by novel disease dynamics in the pandemic era.

## 5. Conclusions

Ultimately, we found that state weather patterns during the influenza epidemic were significantly different from previous years and matching typical winter temperatures and that there was a mismatch between influenza vaccine strains and the strain responsible for the influenza epidemic. Our findings suggest that the seasonality of respiratory viruses may be unexpected and unpredictable during the COVID-19 pandemic. That interruption of typical disease dynamics and predisposition to off-season epidemics may be due to the changing landscape of respiratory virus and pathogen exposure, as well as the impact of climate change on weather patterns. As such, we should expect off-season pathogen outbreaks during the COVID-19 pandemic and be prepared to prevent and treat such outbreaks.

## Figures and Tables

**Figure 1 vaccines-10-00821-f001:**
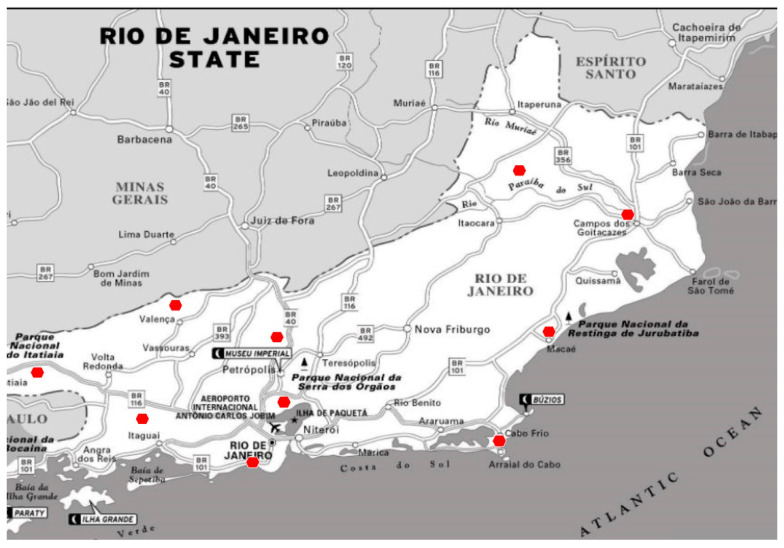
Representative meteorological stations across the State of Rio de Janeiro, Brazil.

**Figure 2 vaccines-10-00821-f002:**
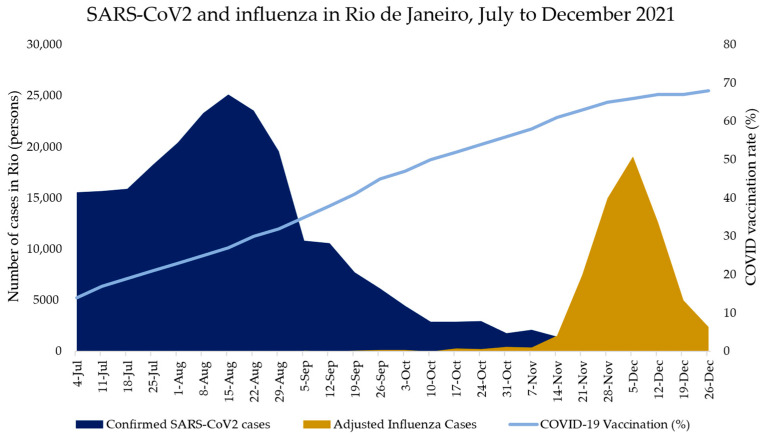
SARS-CoV2 and influenza in Rio de Janeiro, July to December 2021. Confirmed SARS-CoV2 cases and adjusted influenza cases in Rio de Janeiro are represented in blue and yellow bars, respectively. The number of SARS-CoV2 cases and adjusted influenza cases in Rio de Janeiro is described on the left *y*-axis. The COVID-19 vaccination rate in Rio de Janeiro is shown as a light blue line and is described on the right *y*-axis. The *x*-axis represents the weeks from 4 July 2021 to 26 December 2021, in which cases were reported in Rio de Janeiro.

**Figure 3 vaccines-10-00821-f003:**
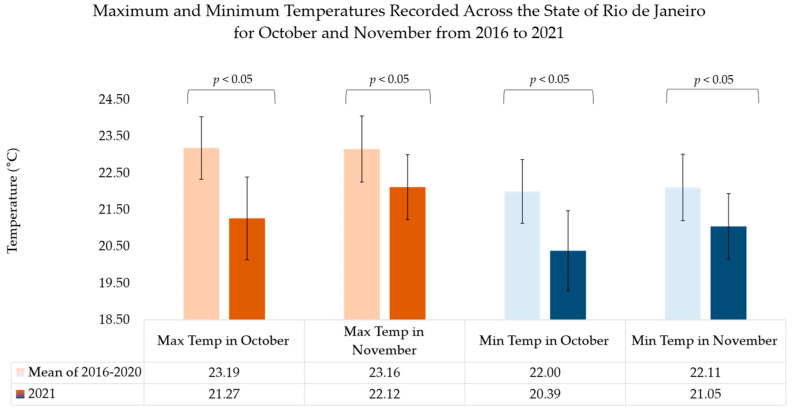
Maximum and minimum temperatures recorded across the State of Rio de Janeiro for October and November from 2016 to 2021. Data from 2016–2020 and 2021 were averaged between ten weather stations covering the State of Rio de Janeiro. The data for the mean of 2016–2020 represent all measurements taken from maximum and minimum temperatures for the ten weather stations in Rio de Janeiro. A Wilcoxon signed rank test was performed comparing 2021 maximum and minimum temperatures for the months of October and November with that of the mean of 2016–2020. Error bars represent standard error. All *p*-values were <0.001.

**Table 1 vaccines-10-00821-t001:** Hospitalizations and cases of influenza, circulating influenza strains, vaccines used and vaccine coverage in the State of Rio de Janeiro, 2016–2021 [12,13,14,15,16,17,18,19,20,21].

Year	Hospitalizations	Cases *	Month(s) with the Highest Number of Cases (S6–8)	Circulating Strain (S1–3,8)	Trivalent Vaccine Strains (S5)	Vaccination Coverage (S4)
2016	2484	191,076	May–June	A/(H1N1)pdm09 **	A/California/(H1N1) pdm09 A/Hong Kong/(H3N2) B/Brisbane	91%
2017	1155	88,846	May–June	A/(H3N2)	A/Michigan/(H1N1)pdm09 A/Hong Kong/(H3N2) B/Brisbane	85%
2018	1898	146,000	May–June	A/(H1N1)pdm09	A/Michigan/(H1N1)pdm09 A/Singapore/(H3N2) B/Phuket	78%
2019	2394	184,154	May–June	A/(H1N1)pdm09	A/Michigan/(H1N1)pdm09 A/Switzerland/(H3N2) B/Victoria	93%
2020	0	NA	NA	NA	A/Brisbane/(H1N1)pdm09 A/South Australia/(H3N2) B/Washington	91%
2021	885	68,077	November	A/Darwin (H3N2)	A/Victoria/(H1N1)pdm09 A/Hong Kong/(H3N2) B/Victoria	58%

* The data for 2016–19 are based on cases of acute respiratory distress syndrome (ARDS) admitted to sentinel hospitals for grave illness attributed to influenza [20,21] with the total number of cases calculated based on CDC methodology (hospitalizations comprise 1.3% of cases). The year 2020 is listed as “Not Applicable (NA)” because the number of influenza cases reported in the study region was exceptionally low due to the COVID-19 pandemic with no hospitalizations reported, so an estimate of the number of total cases is not possible. The year 2021 includes all influenza cases and estimated hospitalizations during the out-of-season influenza epidemic (as per Results). ** “pdm” denotes “pandemic” in reference to the 2009 pandemic influenza strain.

**Table 2 vaccines-10-00821-t002:** Minimum, mean and maximum temperature (°C) in State of Rio de Janeiro, 2016–2021. Monthly temperature data for this period for weather stations in the state were obtained from the Brazilian National Institute of Meteorology [11] (see Methods). For each station, we determined the annual minimum, mean and maximum temperatures, as well as the October and November minimum and maximum temperatures. For each of these variables, we report the mean and standard error of the observations from all of the stations.

Year	Minimum (°C)			Mean (°C)	Maximum (°C)		
	October Only	November Only	Whole Year	Whole Year	October Only	November Only	Whole Year
2016	21.17 (20.2–22.14)	22.04 (21.12–22.96)	21.62 (20.58–22.65)	22.16 (21.13–23.2)	22.42 (21.51–23.34)	23.11 (22.22–24.01)	22.74 (21.71–23.77)
2017	22.33 (21.61–23.06)	22.01 (21.15–22.87)	21.58 (20.68–22.47)	22.15 (21.27–23.04)	23.77 (23.08–24.46)	23.09 (22.23–23.95)	22.76 (21.88–23.65)
2018	21.74 (20.8–22.67)	22.34 (21.36–23.32)	21.07 (20–22.13)	21.71 (20.69–22.73)	22.67 (21.72–23.61)	23.35 (22.36–24.33)	22.15 (21.08–23.22)
2019	22.42 (21.54–23.31)	22.12 (21.23–23.01)	22.1 (21.18–23.03)	22.67 (21.74–23.6)	23.57 (22.69–24.45)	23.11 (22.23–24)	23.27 (22.34–24.2)
2020	23.2 23.07–23.33)	22.81 (22.62–23)	22.14 (21.18–23.03)	22.69 (21.74–23.6)	24.33 (24.21–24.45)	23.91 (23.76–24.05)	23.27 (22.57–23.97)
2021	20.39 (19.3–21.48)	21.05 (20.16–21.94)	21.11 (20.19–22.04)	21.68 (20.76–22.6)	21.27 (20.14–22.39)	22.12 (21.24–23.01)	22.28 (22.57–23.97)

## Data Availability

Publicly available datasets were analyzed in this study. These data can be found here: [http://sistemas.saude.rj.gov.br/tabnetbd/dhx.exe?COVID19/tf_evento_divulgacao.def] (accessed on 2 March 2022), and [http://sistemas.saude.rj.gov.br/tabnetbd/dhx.exe?sivep_gripe/sivep_gripe.def] (accessed on 14 March 2022), [http://tabnet.datasus.gov.br/cgi/dhdat.exe?bd_pni/dpnibr.def] (accessed on 14 March 2022). Data can also be provided upon request.

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
