# Peer review of "Out-of-Season Influenza during a COVID-19 Void in the State of Rio de Janeiro, Brazil: Temperature Matters"

_vaccines, 2022, doi:10.3390/vaccines10050821_

Round 1

Reviewer 1 Report

The work could perhaps be interesting if supported with further information and analysis on detailed and not aggregated data and by considering time series of influenza cases over different years linked with analogue series for relevant determinants like temperature.

Reviewer 2 Report

In this report, the authors sought to evaluate the roles for climate change, poor influenza immunization coverage, and a surge in Omicron infections in the unexpected increase in influenza H3N2 cases in Rio de Janeiro, Brazil in Fall 2021. While the authors do an admirable job in collecting and reporting data related to the number of COVID-19 cases, COVID-19 vaccination rates, number of influenza cases, and temperature trends in Rio de Janeiro, my major issue with the study is that the authors fail to establish a clear, convincing, and statistically supported association between any of these factors and out-of-season outbreaks of influenza. My other comments:

  1. For Figure 2, it is unclear whether the numbers on the y-axis relate to both confirmed SARS-CoV2 cases and influenza cases. The addition of a figure legend providing a clear and concise explanation of the data shown is highly recommended.
  2. For Figure, it is unclear what the bars and error bars represent. How were means calculated? For 2021, does the bar represent means calculated from all meteorological stations? For the 2016-2020, does the bar represent all measurements taken throughout a month by all meteorological stations? Again, I highly recommend the addition of a figure legend to provide the reader with a clear and concise description of what data is shown.
  3. In the Discussion the authors suggest that year-round influenza vaccines could mitigate out-of-season outbreaks. This approach seems implausible given the appearance of new flu strains annually (i.e., antigenic drift) and the extended time needed to manufacture, test, and distribute new vaccines to combat annual epidemics.

Reviewer 3 Report

Comments to the authors
I have read and reviewed the paper “Out of Season Influenza during a COVID-19 Void in the State 2 of Rio de Janeiro, Brazil: Temperature Matters,” very carefully and found that the results of the paper are looking interesting and mathematically correct. The novelty of the results is also good and the presentation of the paper is suitable. I recommend this paper for publication in this journal, but after some corrections. My suggestions are given as follows:
1. The abstract should be concise and more informative. The significant outcome of the study needs to be mentioned.
2. Improving the introduction and discussing some recent literature on Covid-19 disease as the current introduction is not enough. I suggest the following recent works to the authors:
i. Modeling and analysis of COVID-19 epidemics with treatment in fractional derivatives using real data from Pakistan. European Physical Journal Plus, 2020, 135(10):795.
ii. Investigation of Interactions Between COVID-19 and Diabetes with Hereditary Traits Using Real Data: A Case Study in Turkey. Computers in Biology and Medicine 141(3):105044, 2022.
3. The authors should highlight in their abstract as well as in discussion and conclusion part of their paper about possible applications of their results in disease dynamics.
4. The physical meaning of figures must be explained more in the discussion part.
5. The conclusion part of the paper should be included at the end, discussing the significant outcome of the study.
The results are correct. I recommend it for publication after the above-suggested revisions.

Reviewer 4 Report

The present manuscript reports an out of season type A Influenza (H3N2) epidemic occurred in the state of Rio De Janeiro, Brazil in October-November 2021, during the period of low COVID-19 load in the state. The authors assessed the role of temperature and climate change along with immunization against Influenza during this event among the population.

The data presented, however does not clearly identify temperature as the exact cause and correlation of temperature and Influenza cases with the other prevailing factors like social distancing, mask use and Influenza vaccination among the population. Also, there is no clear indication as to how the data is extracted according to the type of population studied.

I would like to suggest the following revisions:

  1. Line number 14: Please clearly denote the months of Delta and Omicron surges in the state.
  2. Line number 13 and 45: Kindly omit opportunistic. Influenza is not characterized under Opportunistic infection.
  3. Line number 31, 34, 38: Kindly specify clearly the months associated with different seasons observed in the state.
  4. Line Number 60: Please specify the time period from which the data was taken for Influenza case numbers.
  5. Line Number 61: Elaborate CDC methodology and how it is employed in the calculation of weekly state influenza case numbers.
  6. Line number 64-82 (Para 2.3 & 2.4): Kindly specify the population age considered for the report. In abstract, the population specified is as adults while the data for SARS CoV-2 and Influenza immunization coverage is specified as individuals above 12 years of age and 6 months of age, respectively.
  7. Line number 98: Please specify from the total cases the number of cases in adult population.
  8. Line Number 103: Kindly denote case fatality rate for COVID-19 in percentage.
  9. Line number 107-108: Kindly specify if the Influenza vaccine coverage mentioned here is for adult population or total state population.
  10. Line number 120-122: Please add the values for mean temperature during epidemic and for the winter months of July and August in a year.
  11. Line number 131-140: Please add the values for number of SARS CoV-2 and Influenza cases recorded during Delta & Omicron surge in the state. Clearly mention the Influenza cases recorded during void and surge of SARS CoV-2 and during Influenza epidemic.
  12. Line number 152-155: Please explain how temperature and weather conditions affects the surge of Influenza virus.
  13. Line number 160: Please explain how the population clustering indoors during winters played a role in spread of Influenza virus and out-of-season epidemic.
  14. Please show the data for minimum and maximum temperatures and average temperature in a year from 2016-2021 in the state of Rio De Janeiro. It is advisable if a detailed table can be added for the same.
  15. Please show the data for Influenza cases recorded from 2016-2021 during seasonal or out of season outbreak, if any. Kindly specify the data according to months. Specify the type of strain circulating during those outbreaks.
  16. Please show the data for vaccination status for Influenza in the preceding 5 years and the type of vaccine used.

Round 2

Reviewer 1 Report

The topic is interesting, but to be scientifically and statistically sound it should be improved and enlarged by more detailed data allowing to build models able to validate the assumptions, that are only descriptive.

In any case, I would suggest to present the paper as a Communication rather than as a research article, due to the aforementioned critical issues.

Moreover, I would ask to assess whether there have been other periods in which a decrease in temperature out of winter season have produced an increase in the cases of influenza and for which amount. This would allow to distinguish better, even though always at a descriptive level, the contributions from Covid respite and climate.

As a second point, I would ask to explicitly specify from the beginning that the conclusions drawn from the study are only based on a descriptive analysis of the data considered and not on a deeper statistical insight into their possible association.  

As far as specific corrections are concerned:

  • page 3 line 103: if the t-test is performed with a sample of only 10 observation, a Wilcoxon test would be required instead
  • page 3 line 117: the CFR is not estimated, from what is said at page 2 line 70 the absolute number of deaths is known and on the opposite the case numbers is estimated from it
  • page 4 line 140: why don't you consider also the cases of influenza that arose during the winter period, even though they should be far less? Is the report not available? In any case, the number of cases should be showed for all years (when data are available) in both winter and out-of-season periods to produce a complete and reliable comparison
  • page 5 line 155: there is an error (17 instead of something else)
  • page 6 line 190: are there are no publications in literature concerning the same phenomenon in other South America states or regions?

Author Response

The topic is interesting, but to be scientifically and statistically sound it should be improved and enlarged by more detailed data allowing to build models able to validate the assumptions, that are only descriptive.

In any case, I would suggest to present the paper as a Communication rather than as a research article, due to the aforementioned critical issues.

Moreover, I would ask to assess whether there have been other periods in which a decrease in temperature out of winter season have produced an increase in the cases of influenza and for which amount. This would allow to distinguish better, even though always at a descriptive level, the contributions from Covid respite and climate.

In response to a prior reviewer request we added Table 1, where we show data for  influenza outbreaks in the five preceding years. All outbreaks occurred between May to June from 2016 to 2019. There was no influenza circulation in 2020 during the time of the COVID-19 pandemic, as per the Rio de Janeiro/ Brazilian health department surveillance data. There were no outbreaks recorded for influenza virus in months outside fall/ winter in the five preceding years before the 2021 influenza outbreak in the state of Rio de Janeiro.   We also added Table 2, where we describe year-round temperatures and minimum and maximum temperatures in October and November from 2016 to 2021.As compared to prior years, the minimum and maximum temperatures for October and November in 2021 were lower. Therefore, we state that lower mean temperatures in October and November 2021 may have contributed to the off-season influenza epidemic. Similarly, we describe the unprecedented and unexpected circumstance where there were no influenza cases reported by the Rio de Janeiro Health Department during the Delta SARS CoV-2 epidemic surge. Following the surge, we saw the off-season influenza epidemic. Therefore, we state that in addition to changing weather patterns, other prevailing competing viruses (e.g., SARS CoV-2) may have contributed to this unique phenomenon.

We have added a sentence to the text (lines 158-161) where we state: In the preceding years, in the state of Rio de Janeiro,  there were no periods in which a decrease in temperature outside of the winter season produced an increase in the cases of influenza. However, the caveat to this observation is, as shown in Table 2, temperatures in the five preceding years were warmer than in 2021.  

As a second point, I would ask to explicitly specify from the beginning that the conclusions drawn from the study are only based on a descriptive analysis of the data considered and not on a deeper statistical insight into their possible association.  

We have added this statement to the introduction in lines 54-55.

We evaluated whether climate change and poor influenza immunization coverage contributed to the genesis of this perfect storm and share findings based on a descriptive analysis of the data considered.

As far as specific corrections are concerned:

  • page 3 line 103: if the t-test is performed with a sample of only 10 observation, a Wilcoxon test would be required instead

We repeated the analysis as requested with a Wilcoxon test. The test still demonstrated a statistically significant difference between the temperatures in October and November and prior years, thus rejecting the null hypothesis that there was no difference between 2021 maximum and minimum temperatures for the months of October and November as compared to the mean of 2016-2020 for the same months.

  • page 3 line 117: the CFR is not estimated, from what is said at page 2 line 70 the absolute number of deaths is known and on the opposite the case numbers is estimated from it

The word “estimated” has been removed. The reviewer is correct, the absolute number of deaths is known, and the case numbers were estimated based on the deaths. The influenza burden was calculated from CDC data from 2018-2019, in which for 29 million symptomatic cases, there were 380,000 hospitalizations and 28,000 deaths due to influenza. 28,000 deaths/29 million cases = 0.1% CFR

  • page 4 line 140: why don't you consider also the cases of influenza that arose during the winter period, even though they should be far less? Is the report not available? In any case, the number of cases should be showed for all years (when data are available) in both winter and out-of-season periods to produce a complete and reliable comparison

Table 1 shows number of cases reported for each year (except 2020, when there were no influenza hospitalizations recorded). 2021 data includes out-of-season epidemic cases and not winter cases because there were no influenza hospitalizations recorded during the winter of 2021 in the state of Rio de Janeiro. All hospitalizations due to respiratory illnesses during the winter of 2021 were due to COVID-19 during the Delta surge. No hospitalizations due to influenza were recorded in the winter months.

  • page 5 line 155: there is an error (17 instead of something else)

The word “of” has been added to this sentence: “The median and interquartile ranges of monthly minimum, mean, and maximum temperatures of 17 weather stations in the state of Rio de Janeiro are reported from 2016 to 2021 in Table 2”

  • page 6 line 190: are there are no publications in literature concerning the same phenomenon in other South America states or regions?

We have not identified any publications in the literature that describe this phenomenon in Brazil or any other South American state or region during this time period. We have a reference quoted for an influenza outbreak in South Africa which was also atypically out of season (Ref 31). 

Reviewer 2 Report

In this report, the authors sought to evaluate the potential impacts of unseasonable temperatures poor influenza immunization coverage, and a surge in Omicron infections in the unexpected increase in influenza H3N2 cases observed in Rio de Janeiro, Brazil in Fall 2021. I thank the authors for their careful consideration of my comments and critiques. The revisions, particularly the extended figure legends and inclusion of Table I, have resulted in an improved manuscript. Though the data shown still do not establish a statistically supported association between any of these factors and out-of-season outbreaks of influenza, the data do stimulate thoughts/additional studies on how traditional disease dynamics have been disrupted due to the pandemic, and potentially climate change. I have some minor comments on the revised manuscript:

  1. The word "significantly" should be removed from line 18. The authors have not performed any correlation coefficient calculations to establish a statistical relationship between weather patterns and influenza cases. Other factors such as vaccine mismatch and a considerable lower than normal flu vaccine coverage rate likely had a greater impact on flu cases in 2021.
  2. line 211: There is clear and ample evidence that individuals can be co-infected with influenza A and SARS-CoV-2. There is little data to suggest that either virus can reduce or enhance infection by the other virus. Thus, the authors need to either remove this statement or provide a better context for their statement.
  3. lines 217-22: The authors raise valid theories as to why cases of respiratory virus infections can increase during cold weather months. These ideas may explain in part why cases of influenza unexpected spiked in Rio de Janeiro between November and December 2021. However, once would have expected cases of other respiratory viruses (especially COVID-19) to have spiked during this time period. The authors should comment on this, particularly as it relates to COVID-19 (i.e., % of Brazilians fully vaccinated against COVID during this time interval; reappearance of COVID cases in Brazil in early January 2022). 

Author Response

  1. The word "significantly" should be removed from line 18. The authors have not performed any correlation coefficient calculations to establish a statistical relationship between weather patterns and influenza cases. Other factors such as vaccine mismatch and a considerable lower than normal flu vaccine coverage rate likely had a greater impact on flu cases in 2021.

The word “significantly” has been removed from line 18.

  1. line 211: There is clear and ample evidence that individuals can be co-infected with influenza A and SARS-CoV-2. There is little data to suggest that either virus can reduce or enhance infection by the other virus. Thus, the authors need to either remove this statement or provide a better context for their statement.

We have removed line 211.

  1. lines 217-22: The authors raise valid theories as to why cases of respiratory virus infections can increase during cold weather months. These ideas may explain in part why cases of influenza unexpected spiked in Rio de Janeiro between November and December 2021. However, once would have expected cases of other respiratory viruses (especially COVID-19) to have spiked during this time period. The authors should comment on this, particularly as it relates to COVID-19 (i.e., % of Brazilians fully vaccinated against COVID during this time interval; reappearance of COVID cases in Brazil in early January 2022). 

We added a comment hypothesizing why SARS CoV-2 cases did not rise during the unexpectedly cold weather months. Please see lines 224-233:

“In contrast, by mid-November, the COVID vaccination rate in RJ exceeded 70 percent (as shown in Figure 2), which may have played a role in protecting the RJ population from the Delta strain. In addition, a considerable proportion of the population acquired COVID-19 during the Delta surge before November 2021. Thus, both vaccination and recent natural immunity likely contributed to a lack of resurgence of COVID-19 cases.”

Reviewer 4 Report

As per the suggestion, the authors have successfully revised and updated the manuscript.

Author Response

Thank you for your consideration and for your time.